# Seroprevalence of Equine Influenza Virus Antibodies in Horses from Four Localities in Colombia

**DOI:** 10.3390/v17070999

**Published:** 2025-07-16

**Authors:** Juliana Gonzalez-Obando, Jeiczon Jaimes-Dueñez, Angélica Zuluaga-Cabrera, Jorge E. Forero, Andrés Diaz, Carlos Rojas-Arbeláez, Julian Ruiz-Saenz

**Affiliations:** 1Grupo de Investigación en Ciencias Animales—GRICA, Facultad de Medicina Veterinaria y Zootecnia, Universidad Cooperativa de Colombia, Bucaramanga 680002, Colombia; juliana.gonzalezo@udea.edu.co (J.G.-O.); jeiczon.jaimes@campusucc.edu.co (J.J.-D.); 2Grupo de Epidemiología, Universidad de Antioquia, Medellín 050010, Colombia; carlos.rojas@udea.edu.co; 3Grupo de Investigación GISCA, Facultad de Medicina Veterinaria y Zootecnia, Fundación Universitaria Vision de las Américas, Medellín 050031, Colombia; angelica.zuluagac@uam.edu.co; 4Grupo de Investigación en Microbiología Ambiental, Escuela de Microbiología, Universidad de Antioquia, Medellín 050010, Colombia; jorge.forero@udea.edu.co; 5Pig Improvement Company, Hendersonville, TN 37075, USA; andres.diaz@genusplc.com

**Keywords:** epidemiology, vaccination, Colombia, horses, infection, influenza A virus

## Abstract

Equine influenza is a highly contagious disease caused by the equine influenza virus (EIV). The occurrence of EIV outbreaks in America is associated with low levels of vaccination coverage. In Colombia, no seroprevalence evaluation has been carried out to estimate the distribution of the virus within the country. Our aim was to perform a sero-epidemiological survey of equine influenza infections and to identify associated risk factors in horses from four departments of Colombia. Serological testing was carried out by using an ELISA for the detection of IgG antibodies against the influenza A virus. The evaluation of epidemiological variables, clinical manifestations, and vaccination history was carried out through the application of a data collection instrument. Among the 385 horses analyzed, 27% of the samples tested positive, with a higher prevalence in Study 1 from horses with respiratory symptoms (40.4%) than in Study 2 from horses without clinical signs (16.1%). Only horses housed in stables had higher odds of testing positive. The study also revealed that unvaccinated horses were 68% less likely to test positive than vaccinated horses were. This research highlights a significant gap in vaccination coverage and the presence of antibodies even in asymptomatic horses. Management factors such as activity type and housing should be considered when strategies for EIV prevention are developed.

## 1. Introduction

EIV, primarily the H3N8 subtype, has caused outbreaks worldwide [1]. EIV outbreaks in the Americas have been reported in Colombia [2,3], Ecuador [2], Argentina [4,5], the United States [6], Canada [7], Mexico [8], Chile [9,10], and Uruguay [2,6,11]. The primary impact of EIV is on naive or nonvaccinated populations [12,13,14], in which symptoms appear within 3 to 10 days after infection. However, some horses can be asymptomatic after infection [15,16]. Vaccination represents the most effective strategy to control EIV infections because it reduces the severity of clinical symptoms and transmission [13,17,18,19]. Nevertheless, the capacity of EIV to continuously accumulate mutations allows the virus to escape immunity and contributes to the success of EIV as a viral pathogen. Additionally, the protective effect of the EIV vaccine wanes over time, favoring subclinical infections, transmission, and virus evolution [17,20].

In Colombia, EIV vaccination is required to move horses between premises [13,21]. Since 2010, the vaccines used in Colombia have targeted Florida Clades (FCs) 1 and 2, as outlined by the World Organization for Animal Health (WOAH) expert panel [22] The most widely used vaccines in the country are inactivated ones, such as Prequenza™; accordingly, animals must have a complete and up-to-date vaccination schedule at the time of mobilization [23]. Despite vaccination, increased movement of horses, both nationally and internationally, particularly those intended for competitive events, is correlated with EIV transmission and outbreaks in Latin America [4,20]. Additionally, analyzing antibody frequency in relation to vaccination schedules and vaccine types can help elucidate the relationship between vaccination strategies and effective immune responses [15,24,25]. The detection of antibodies following vaccination or natural infection can occur anywhere from 10 days to 12 months after exposure to EIV; however, there is a significant decline in antibody levels 6 months after vaccine administration [26,27]. The serological diagnosis of EIV is crucial from both clinical and epidemiological perspectives for understanding outbreaks [9,10,11].

The epidemiology of EIV is not completely understood because EIV transmission and outbreaks are associated with horse movement [2], a lack of vaccination [10], outdated vaccines [18], and antigenic drift [28,29]. In Latin America, studying the distribution of EIV antibodies allows for the understanding of geographical exposure over time and the assessment of exposure within a population. Furthermore, serological investigations of EIVs provide a means to assess risk factors for susceptible populations in specific areas and then design better health interventions to minimize the impact of EIVs on equine health and performance. Our aim was to perform a sero-epidemiological survey of equine influenza infections and to identify associated risk factors in horses from four departments of Colombia.

## 2. Materials and Methods

Three hundred eighty-five serum samples conveniently collected from four localities in Colombia (Antioquia (ANT), Cundinamarca (CUN), Santander (SAN), and Arauca (ARA) (Figure 1)) were used in this study. ANT, CUN, and SAN were selected on the basis of their high levels of equine activity and the presence of a substantial population of sport and competition horses [30]. ARA was selected because of its geographical location in the Eastern Plains, a region of strategic importance where equestrian activities and equine competitions are also prominent [31]. The serological status of these horses in response to EIV and its possible associations with different demographic (e.g., breed, age, and location), zootechnical (e.g., equestrian or other purpose), other use (working or breeding horses), or housing (stable vs. pasture) variables were estimated. The variables included in the analysis were obtained from the following data collection instrument (Appendix A). Samples were obtained from two different studies and collected at different time points. Study 1 was collected from equines with respiratory symptoms between 2020 and 2023 from ANT (*n* = 121) and CUN (*n* = 67). Samples from Study 2 (*n* = 197) were collected between 2020 and 2021 from horses with or without respiratory symptoms in the SAN (*n* = 114) and ARA (*n* = 83) localities. Historical data for each sample were reviewed to establish the variables of interest in this study (Table 1). Both samples were conveniently selected according to the information on symptomatic individuals (Study 1) and the possibility of access to the samples (Study 2), and adhered to both international and Colombian ethical guidelines for biomedical research involving animals, as stipulated by bioethical and animal research regulations. The studies were reviewed and approved by the Ethics Committee for Animal Experimentation at Universidad Cooperativa de Colombia in Bucaramanga (Minute number 044-2018) and (Minute number 001-2021). Written informed consent was also obtained from all owners of the animals involved in the study.

### 2.1. Sample Collection and Laboratory Analysis

A total of 3 mL of blood was collected for subsequent separation of the serum in the laboratory. Serum samples were separated at 2600 rpm for 15 min and stored at −80 °C until serological antibody analysis. The samples were analyzed via an IDEXX AI Multi-Screen-Influenza A Ab ELISA Test^®^ (IDEXX Laboratories, Inc., Westbrook, ME, USA). This method detects IgG antibodies in serum directed against the nucleoprotein (NP) of the influenza A virus, with a specificity and sensitivity of 95.5% and 99.6%, respectively [32]. The IDEXX ELISA test has been used for the detection of antibodies following infection, as well as for assessing the presence of antibodies induced by vaccination [33]. This test has the capacity to detect antibodies from 7 to 10 days after their appearance and can remain effective in detecting them for up to 10 months. Briefly, 100 µL of serum was taken from each sample for subsequent dilution (1:10), which was added to a plate previously coated with the animal influenza virus NP antigen; two wells were used for each of the controls and for the horse samples. Three washes were subsequently performed, and an anti-NP antibody conjugated to horseradish peroxidase was added to the mixture, after which a substrate was added to the mixture, and a stop solution was added to stop the reaction. The procedures were performed in accordance with the manufacturer’s instructions.

The results were evaluated via spectrophotometry using a Multiskan™ ^®^ FC Microplate Photometer (Thermo Scientific, Inc., Waltham, MA, USA) at 650 nm and interpreted in reference to the negative and positive reaction controls supplied with the Kit Samples. A positive and negative control included in the ELISA kit were included in the study. A sample-to-negative ratio (S/N) lower than 0.60 was considered positive for influenza A virus antibodies.

### 2.2. Statistical Analysis

Tabular methods were first used to estimate the odds of a positive sample given the exposure variables (demographic, zootechnical, and housing) (Table 1). Additionally, for the samples collected in Study 2, the proportions of positive and negative samples were compared between animals with or without clinical signs. Then, a logistic regression model was set up in a backward step to adjust the estimations, including all variables with a *p* value lower than 0.25 from the crude analysis, and for the adjusted analysis, those that were not significant for the model with *p* > 0.05 were removed. For simplicity, locality was divided into two dichotomies: Antioquia and others. The analysis was conducted via the licensed statistical application SPSS^®^ v.21 (IBM, Armonk, NY, USA), and confidence intervals were estimated via Epiinfo^®^ software (Centers for Disease Control and Prevention (CDC), Atlanta, GA, USA).

## 3. Results

### 3.1. Descriptive Analysis

The total number of equine samples from ANT, SAN, ARA, and CUN was 121 (31.4%), 114 (29.6%), 83 (21.6%), and 67 (17.4%), respectively (Figure 1). Overall, 27% of the samples (104 out of 385; 95% CI: 23.50, 32.47) were ELISA positive for EIV. The proportions of positive samples were significantly different by study (*p* < 0.01), locality (*p* < 0.001), activity (*p* = 0.002) and shelter (*p* = 0.000) but not significantly different by sex (*p* = 0.794); Cramer’s V = 0.13 or age (*p* = 0.230); and Cramer’s V = 0.061 (Table 1). The proportion of positive samples was greater in Study 1 (73 out of 188; 40.41% (95% CI: 33.40, 47.42)) than in Study 2 (31 out of 197; 16.13% (95% CI: 10.99, 21.26)) (*p* < 0.001).

### 3.2. Multivariate Analysis

The crude odds of a positive sample were two times greater for samples from horses used for equestrian activities than for samples from horses used for other activities (work and breeding horses, OR: 2.02 (95% CI: 1.25, 3.28; *p* = 0.02)). Additionally, the crude odds of a positive sample were 8.4 times greater for animals housed in stable shelters than for those housed in mixed shelters (pasture/stable), OR: 8.4 (95% CI: 4.55, 15.42: *p* < 0.001). However, the odds of testing positive were 9.9 times greater for animals housed in stable shelters than for those housed in mixed shelters (pasture or pasture/stable shelter, OR: 9.91 (95% CI: 4.83, 20.31) *p* < 0.001) after adjusting for activity and locality (Table 2). However, activity or locality were not statistically significant after adjusting for shelter, indicating that the statistical association between positive samples and activity or locality was not significant (*p* > 0.05) when housing was taken into account to measure the association. The logistic regression model included the following variables: shelter, activity, and locality (*p* < 0.25). Additionally, in the regression, the probability associated with the intercept was approximately 0.270 (27.01%).

### 3.3. Distribution of ELISA Results by Clinical Signature and Vaccination History in Horses from Study 1

The most frequent clinical signs observed in the animals from Study 1 were nasal discharge (99%, 181/188), cough (87%, 155/188), and fever (30%, 48/188). No associations were found between the presence or absence of clinical signs and the ELISA results. However, the odds of testing positive by ELISA for EIV in horses within Study 1 were 68% lower in animals without a history of the EIV vaccine than in those with a vaccine record (OR: 0.32, *p* = 0.01 95% CI: 0.35, *p* = 0.796). Among the vaccinated individuals, 79.2% (19/24) had received their last vaccination more than 6 months prior, whereas only 20.8% (5/24) had been vaccinated within the last 6 months.

## 4. Discussion

This is the first study conducted in Colombia to assess the exposure of horses to EIV antigens and their associations with horses, shelters, localities, and activities. Our results indicate that horse shelter is statistically associated with exposure to EIV antigens and that activity and locality are not relevant when the shelter is considered. These results are important because understanding the risk factors associated with EIV exposure could help prevent virus infection and transmission.

Given these findings, it is essential to consider the mechanisms through which EIV spreads in equine environments. The transmission of EIV poses a significant challenge, particularly because of its airborne circulation and persistence on fomites. A study conducted in the United States demonstrated that the viral RNA of the EIV can be detected in the air before horses exhibit clinical signs, highlighting the risk of infecting healthy individuals in enclosed spaces [34]. Additionally, the virus can remain viable on surfaces for up to 3 days, enabling indirect transmission through contaminated materials [35].

For this reason, the type of shelter may be a key component of EIV epidemiology and transmission because sick horses may spread the disease in close proximity or through fomites. EIV spreads rapidly among horses that live in stables because large amounts of virus are expelled during coughing episodes, which results in a greater distribution of the virus [36,37]; for this reason, stables may be related to the constant circulation of influenza in the environment [34]. The stabling of horses is a practice that has gained popularity in breeding farms and among competition or sport horses over time. Some of the reasons why caretakers prefer this type of shelter include the ease of reducing competition for food, ease of nutritional management, need to conserve pastures, and greater control over water intake [38]. However, this practice may increase the risk of EIV infection because these types of horses are housed in closer proximity to each other [36,37].

Considering this transmission dynamic, biosecurity measures play a crucial role in limiting the spread of the virus. Strategies such as isolating infected animals, controlling equine movement, and frequently disinfecting shelters have proven effective [39]. However, the implementation of these measures varies significantly depending on the available resources and the awareness of stable managers [39,40,41]. In addition, a better understanding of the associations between housing conditions and EIV transmission may help reduce the spread of EIVs or other respiratory viruses in high-risk settings. Moreover, it is important to emphasize that transmission events can significantly alter the genetic composition of a viral population, which may have significant implications for virus evolution and for living in stables and the circulation of the virus [32].

In terms of biosecurity measures, the adoption of strict biosecurity guidelines significantly reduces the spread of the virus. An example of this occurred during the EIV outbreak in Great Britain in 2019 [42]. In particular, controlling horse movement is essential, especially at equestrian events and competitions, where the high concentration of horses facilitates viral transmission within a region or nation. To ensure the effectiveness of these strategies, biosecurity measures must also be implemented consistently and continuously in high-density and enclosed equine shelters, such as stables [39].

Some recommended practices for the prevention and control of diseases in equine shelters include changing clothing and footwear before entering the premises, ensuring the use of disinfected overalls and boots, and cleaning and disinfecting equipment used in horse handling [43]. Separating clean and dirty areas is also recommended to minimize disease spread, including the disinfection of vehicles entering the property, particularly those used for horse transportation. Products related to disinfection, such as oxidizing and phenolic-based disinfectants, are recommended. Various studies have demonstrated that these disinfectants are highly effective at eliminating EIV on surfaces, even under conditions of high viral loads, such as cases with concentrations up to 6.4 log10 EID_50_ [44]. Furthermore, ethanol, 0.5% chlorhexidine solution, and 70% isopropanol have been shown to effectively inactivate the virus, providing additional options for implementing more effective biosecurity measures in stables [45]. These measures are essential for reducing the risk of pathogen transmission in equestrian environments [46,47].

Future research will be needed to assess the presence of antibodies against EIV in individuals housed in stable environments and not involved in events to determine whether living in such environments is the sole factor associated with the presence of antibodies against the virus. This approach could provide valuable insights into the relationship between equine housing conditions and their immune response to EIV and vaccines. Furthermore, distinguishing between antibodies induced by vaccination and those resulting from natural infection would be beneficial.

In terms of serology frequency, 27% of the horses tested by ELISA were positive for influenza A virus antibodies. This result is consistent with a study conducted in Mexico, where 25% of horses tested were also seropositive using a nucleoprotein-based ELISA [8]. Although the Mexican study included other species and additional viruses, the findings in horses are directly comparable, particularly since the detected antibodies were later confirmed as specific to the H3N8 subtype using HI and NI assays. Likewise, in our study, the presence of H3N8 in the same population was confirmed by PCR in symptomatic animals [13]. Additionally, a higher proportion of ELISA-positive individuals was observed in Antioquia, which may be associated with the timing of sample collection during an outbreak, and the larger number of horses sampled in this department (*n* = 121; 31.4%).

Our findings revealed that a greater percentage of positive animals is likely related to the frequency of antibodies in response to natural exposure to the EIV. The frequency of postinfection antibodies may persist for up to 6 months; however, this immunity tends to decline over time [15,48]. This decline is correlated with the observation that a greater number of individuals exhibit antibodies following a recent or during the outbreak. In contrast, postvaccination antibody levels were lower than expected, which could suggest deficiencies in the administration of vaccination protocols or the effectiveness of the vaccination regimen used. Importantly, when vaccination protocols are properly followed, vaccine-induced antibodies should remain detectable for up to 12 months following the booster dose [23]. This reduction in antibody levels could indicate deficiencies in the implementation of the vaccination regimen, potentially compromising the efficacy of the immunization.

Antibodies against EIV in 27% (104/385) of the asymptomatic horses in Study 1 were present after contact with the virus in the previous months. The identification of antibodies against EIV in asymptomatic individuals is essential for epidemiological surveillance and infection prevention, as reported by Cullinane [28,29]. This surveillance is vital, as it enables the control of outbreaks associated with asymptomatic individuals, who have the potential to transmit the virus to more susceptible populations [49]. However, most likely, and according to the ELISA results, the antibodies are likely to be produced after a natural infection. The study of antibodies in asymptomatic individuals provides a valuable opportunity to estimate specific risks within these populations by integrating variables related to management practices and individual factors in our country. Furthermore, this approach would allow for a more precise analysis of herd immunity within these individuals without apparent symptoms. Such epidemiological studies are crucial for understanding the dynamics of infection in asymptomatic individuals, which could contribute to the development of more effective control and prevention strategies.

The relationship between the presence of antibodies and age has been reported in other studies within the region. One such study conducted in Brazil (2020) revealed a statistically significant difference (*p* = 0.001) in the presence of antibodies between horses younger than 5 years and those aged 5–14 years, with the older group exhibiting higher antibody levels [18,50]. In our study, we observed that the proportion of antibodies was higher in individuals over 10 years of age, possibly because younger individuals tend to have lower antibody levels than older individuals do, increasing their susceptibility to infections. This difference was identified in the crude model; however, no statistically significant difference was found. A possible explanation for the absence of significant differences, which was observed in the study conducted in Brazil, could be related to the sample size in that study and its geographic distribution across 10 states. In this context, future studies with larger sample sizes are recommended, as this could not only increase the observed frequency in equines over 10 years of age, as seen in our study, but also reveal statistically significant differences that contribute to a better understanding of the dynamics studied. Additionally, it is suggested to examine whether other studies have adjusted for housing conditions.

We did not find a significant difference between the ELISA results and activity in the adjusted model. Nevertheless, individuals who attended competitions as well as those who participated in activities such as recreation or competition had a greater frequency of antibodies; this finding may be related to the fact that this population is in contact with other infected individuals during these events, increasing the likelihood of obtaining the infection naturally and subsequent presentation of antibodies [18,35,51]. However, in our research, we did not find a statistically significant difference after applying the logistic regression model. Nevertheless, in our logistic regression analysis, no significant relationship was found between the presence of antibodies and participation in sports activities among individuals. This result could be related to the heterogeneity of the sample or the sample size, factors that may have influenced the ability of the analysis to detect a true association.

A lack of sufficient vaccine coverage or failures in the application of the vaccination scheme can lead to increased EIV transmission [10,11]. In this study, only 12.8% of the sampled horses had received the influenza vaccine, showing how limited vaccination efforts are in the country. In Colombia, influenza vaccination is not mandatory and is usually limited to horses involved in sports or recreational events. Among symptomatic horses (Study 1), 30% of those with antibodies did not have a complete vaccination schedule and presented clinical signs such as coughing, nasal discharge, fever, and respiratory distress similar to what has been reported in other countries in the region, such as Argentina, Ecuador, Chile, and Uruguay [10,11]. EIV vaccination has a significant beneficial effect on animal welfare and reduces economic losses. This vaccination schedule should begin at 6 months of age, and the primary vaccination consists of two immunizations with an interval of 4-6 weeks and a third dose 5-6 months after the last immunization, followed by a subsequent annual booster immunization. Similarly, mares do so 2 weeks before giving birth to avoid infection in foals and respiratory complications [52,53].

With respect to vaccination, 75% (12/16) of the animals that tested positive for antibodies had received one vaccine within 1 year. These findings suggest that these individuals may possess only partial antibody levels associated with their recent vaccination. However, they could still be at risk of infection, as the antibodies elicited by inactivated vaccines typically have a half-life of approximately 6 months [54,55]. Furthermore, in individuals who, despite receiving vaccinations as stipulated by national legislation and product labels, still experience the disease and test positive by PCR [13], the type of vaccine used may be a contributing factor. Specifically, inactivated vaccines, which are used in 100% of vaccinated horses, generally induce a weaker immune response and lower levels of humoral immunity than recombinant vaccines do [17,54,56].

With respect to the postvaccination interval and its relationship with serological positivity, our results are consistent with previous evidence suggesting that the antibody levels elicited by equine influenza vaccination progressively decrease after the first few months postvaccination (3–6 months). Colgate et al. (2023) demonstrated that horses vaccinated more than 6 months prior are at higher risk of infection and more likely to develop clinical signs, particularly in the context of active viral circulation [57]. In addition, other studies demonstrated that protection against equine influenza decreases 3 months after vaccine administration, increasing the susceptibility of horses to infection [27,58]. Given the progressive decline in postvaccination immunity and the high transmissibility of the virus, the implementation of complementary biosecurity measures is essential. Key strategies include isolating symptomatic animals, controlling horse movement, disinfecting facilities and personnel, maintaining up-to-date vaccination schedules, and applying quarantine to newly introduced horses.

These findings support the interpretation that the seropositivity observed in horses vaccinated more than 6 months before sampling in our study may be associated with recent natural exposure, which is consistent with the presence of acute respiratory symptoms in all the sampled animals. In Colombia, vaccination is required for the movement of horses. According to national regulations, all equines over 6 months of age must be vaccinated against EIV with products duly registered by the Colombian Agricultural Institute to be eligible for transport; the vaccination is valid for a year [59]. There are no national statistics on overall vaccination coverage. However, the literature recommends more frequent booster vaccinations every 6 months for competitive horses due to the elevated risk of infection. Furthermore, it is advised to administer an additional booster 1 to 2 weeks prior to an event if there is an increased risk of infection due to exposure to other populations and stressors such as transportation [60]. Relatedly, a lack of proper vaccination, which is linked to inadequate awareness of the importance of a complete vaccination schedule, has been documented in other studies worldwide and in South America [2,13]. This issue is often associated with insufficient knowledge regarding the economic impact of the disease and the cost-benefit analysis of comprehensive vaccination. Effective vaccination not only offers individual protection but also helps reduce the frequency and severity of EIV within equine populations [1,50,61].

Although the findings of our study are important for a better understanding of the epidemiology of EIV, we also recognize its limitations. First, samples were conveniently selected from two previous studies and not randomly selected to represent a population; hence, the results are biased by sample selection Studies 1 and 2; in general, the equines included in Study 1, originating from the departments of ANT and CUN and primarily intended for sporting and competitive purposes, also presented a relatively high density of equine populations. This may be considered a sampling bias and could have influenced the differences in antibody prevalence observed between Studies 1 and 2. However, our findings are in agreement with those of previous studies and support the biological plausibility that certain shelter conditions or interactions may increase the risk of EIV exposure. Second, the ELISA test used for measurement does not distinguish between antibodies induced by natural infection and those induced by vaccines [62]. However, our results indicate that vaccinated individuals were not more likely to test positive via ELISA, suggesting that antibodies induced by vaccines in these horses may have waned, hence testing negative via ELISA. The seropositivity detected in our study likely reflects exposure to the equine influenza virus (EIV) subtype H3N8, which has been the only subtype reported in equine outbreaks in Colombia and throughout Latin America in the past three decades. This interpretation is further supported by molecular confirmation of H3N8 infection through RT-PCR and next-generation sequencing in symptomatic horses from the same population [13]. However, since the ELISA test used (IDEXX AI Multi-Screen-Influenza A Ab ELISA Test^®^) detects antibodies against the nucleoprotein of influenza A viruses, it does not allow subtype differentiation. Therefore, we cannot exclude the theoretical possibility of other subtypes, although there is no epidemiological evidence supporting the circulation of other influenza A subtypes in the region. The subtype that has been reported in the region in recent decades is H3N8 [2,14,17]. We acknowledge this as a limitation of the study, and future research incorporating subtype-specific serological assays, such as hemagglutination inhibition (HI) or single radial hemolysis (SRH), would provide more definitive confirmation. More studies on the molecular detection of EIV are needed to clarify the role of shelter and other variables in the transmission of EIV within and between horses. Moreover, the NS1 protein in EIVs is highly conserved and is produced only in response to virus replication. Therefore, future studies could use an NS1 ELISA-based test to differentiate between antibodies induced by natural infection or antibodies induced by EIV vaccines, as suggested by [61,63,64,65].

## 5. Conclusions

Our research revealed an association between stable shelters and a higher likelihood of ELISA positivity after controlling for the effects of activity and locality, which may be associated with fomites, limited air circulation, and close proximity to infected horses. These conditions could facilitate greater viral dispersion and greater exposure to EIV in these populations. These findings underscore the importance of the environment in the epidemiological triad (host-pathogen-environment) of infectious diseases, which is crucial for designing health interventions to minimize the negative impact of EIV in horses.

## Figures and Tables

**Figure 1 viruses-17-00999-f001:**
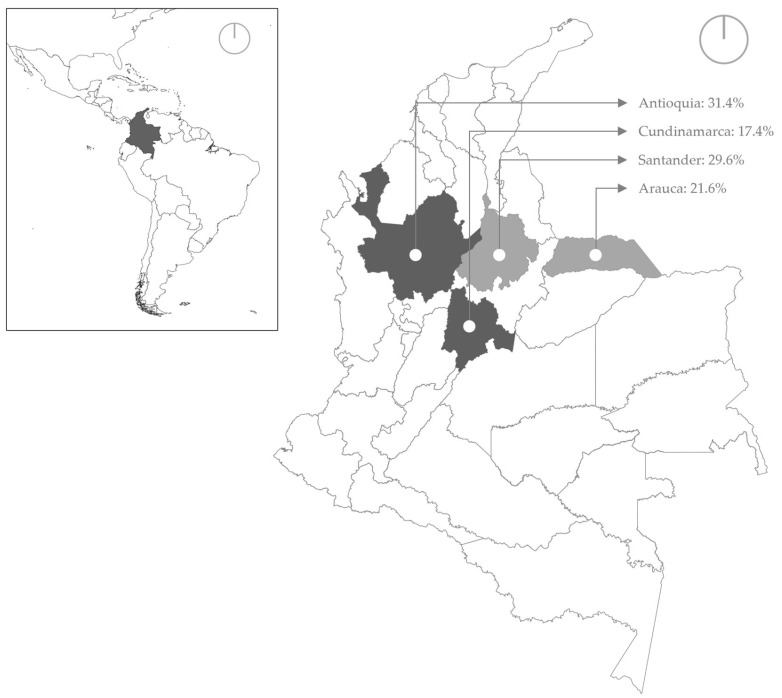
The localities sampled are represented in dark gray for those that included equines in Study 1 and in light gray for those that included equines in Study 2.

**Table 1 viruses-17-00999-t001:** Seropositive samples for equine influenza virus.

Variable	Category	Positive		Total
Study	1	73	(38.80%)	188
	2	31	(15.70%)	197
Locality *	ANT	42	(34.70%)	121
	CUN	31	(46.20%)	67
	ARA	10	(12.00%)	83
	SAN	21	(18.40%)	114
Sex	Male	40	(27.80%)	144
	Female	64	(26.60%)	241
Age	0–2.5	25	(26.60%)	94
	2.5–9.9	59	(25.50%)	231
	<10	20	(33.30%)	60
Activity *	Equestrian	73	(32.60%)	224
	Others	31	(19.30%)	161
Shelter *	Stable	90	(42.50%)	212
	Open air Shelters	14	(8.10%)	173

* Chi-square *p* value < 0.05.

**Table 2 viruses-17-00999-t002:** Multivariate analysis to estimate the odds of positive horses.

Variables	OR	95% CI		*p*-Value	Adjusted OR	95% CI		*p*-Value
Shelter	8.37	4.55	15.42	0.000	9.91	4.83	20.31	<0.001 *
Activity	2.02	1.25	3.28	0.020	0.89	0.50	1.61	0.005
Locality	0.29	0.18	0.47	0.000	0.73	0.42	1.27	0.654

Abbreviations: OR, odd ratio; CI, confidence interval. * Statistically significant.

## Data Availability

The original contributions presented in the study are included in the article, and further inquiries can be directed to the corresponding author. The data that support the findings of this study are openly available at https://doi.org/10.6084/m9.figshare.28755104.v1.

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
