# Peer review of "Seroprevalence of Equine Influenza Virus Antibodies in Horses from Four Localities in Colombia"

_viruses, 2025, doi:10.3390/v17070999_

Round 1
Reviewer 1 Report
Comments and Suggestions for Authors
The authors reported a high seroprevalence of Equine Influenza Virus (EIV) in Colombia. They concluded that horses involved in equestrian activities, particularly those kept in stables, had a higher likelihood of testing positive for the virus. Furthermore, the study found that unvaccinated horses were 68% less likely to test positive for the virus compared to vaccinated horses. The paper could be improved by addressing the suggestions provided, especially in the methodology and discussion sections.
I do not view an overall seroprevalence of 27% as high, even with the 40.4% seroprevalence reported in Study 2. The author should consider revising the title.
Although I understand the authors are limited by word count, I recommend that they include the study design, particularly the administration of the questionnaire, in the abstract.
In the supplementary data, the influenza vaccination history was featured; however, I noticed that no details were presented in the tables. This information is crucial to include in the manuscript to have a fair idea of horses that developed antibodies, possibly due to natural infection, and those that did as a result of vaccination.
The study objectives need to be aligned, as the statement in the abstract differs from the one presented in the introduction section.
The main limitation of the study lies in the authors' assumption that all ELISA-positive samples for influenza A exclusively contained antibodies to equine influenza virus (EIV). According to IDEXX, the manufacturer of the testing kit, it is a multispecies assay. Therefore, it is unclear how the authors concluded that all positive samples specifically represented EIV antibodies without any supporting data for confirmation. Additional testing is necessary to determine whether the ELISA-positive samples indeed contain antibodies developed against EIV. Although the authors acknowledged this as a limitation of their study, this acknowledgement alone does not justify their assumption.
The authors should provide a background on the study locations and justify their choice of the locations.
A sentence or two on EIV vaccination coverage in Colombia should be included to provide better context for the discussion.
Minor comments for the authors.
In the keyword, "influenza virus" should be written as "influenza A virus". Additionally, "vaccination" as "infection". The author should consider including "horses".
To ensure consistency, include the numerical figure beside the percentages where applicable.
Line 75: Delete "other purpose" as it was repeated.
Line 107: In this context, the correct word to use instead of "sensitized" is "coated."
Line 108-110: I believe this was performed using the manufacturer's reagents and chemicals. It will be important to refer to them as such.
Line 117: change "viral" to "virus"
Line 233: Delete "positive"
Comments on the Quality of English Language
The manuscript would greatly benefit from comprehensive editing of language, grammar, punctuation, and sentence structure. For example, to ensure consistency, use a full stop instead of a comma when writing fractions.
Author Response
The authors reported a high seroprevalence of Equine Influenza Virus (EIV) in Colombia. They concluded that horses involved in equestrian activities, particularly those kept in stables, had a higher likelihood of testing positive for the virus. Furthermore, the study found that unvaccinated horses were 68% less likely to test positive for the virus compared to vaccinated horses. The paper could be improved by addressing the suggestions provided, especially in the methodology and discussion sections.
- I do not view an overall seroprevalence of 27% as high, even with the 40.4% seroprevalence reported in Study 2. The author should consider revising the title.
Response: Thank you for the recommendation. We changed the title of the article, by “Seroprevalence of equine influenza virus antibodies in horses from four localities in Colombia” (lines 1-2)
- Although I understand the authors are limited by word count, I recommend that they include the study design, particularly the administration of the questionnaire, in the abstract.
Response: We include in the abstract information related with the questionary (lines 24-26)
- In the supplementary data, the influenza vaccination history was featured; however, I noticed that no details were presented in the tables. This information is crucial to include in the manuscript to have a fair idea of horses that developed antibodies, possibly due to natural infection, and those that did as a result of vaccination.
Response: We included information on the time elapsed since the administration of the last vaccine dose; this information was incorporated in results (lines 173-175) and discussion (321-330) sections.
- The study objectives need to be aligned, as the statement in the abstract differs from the one presented in the introduction section.
Response: Thank you for your review. We adjusted the objective in the Introduction (lines 71-72) to align it with the one presented in the Abstract (lines 21-23).
- The main limitation of the study lies in the authors' assumption that all ELISA-positive samples for influenza A exclusively contained antibodies to equine influenza virus (EIV). According to IDEXX, the manufacturer of the testing kit, it is a multispecies assay. Allrefore, it is unclear how the authors concluded that all positive samples specifically represented EIV antibodies without any supporting data for confirmation. Additional testing is necessary to determine whether the ELISA-positive samples indeed contain antibodies developed against EIV. Although the authors acknowledged this as a limitation of their study, this acknowledgement alone does not justify their assumption.
Response: We thank the reviewer for their observation. We understand the importance of confirming the specificity of the serological results, given that the IDEXX ELISA kit we used is a multispecies assay that detects antibodies against influenza A virus in general. To address this limitation regarding antibody specificity and to strengthen our conclusions, we conducted complementary molecular analyses using RT-PCR and sequencing on selected clinical samples, in which we specifically identified the H3N8 subtype, which is characteristic of equine influenza. Furthermore, in Colombia, only the H3N8 subtype has been reported to circulate in equines, and no other influenza A subtypes have been detected. This information has been clarified in the Discussion section (lines 368-371)
- The authors should provide a background on the study locations and justify their choice of the locations.
Response: We thank the reviewer for their observation. Antioquia, Cundinamarca, and Santander were selected because they are among the departments with the highest equestrian activity in Colombia and host a high concentration of competition horses. Arauca was included due to its location in the Eastern Plains, a region where equestrian and competition activities are also highly relevant. The selection of these areas aimed to cover both regions with a high density of sport horses and strategically important zones for the national equestrian dynamic. This information has been included in lines (77-81).
- A sentence or two on EIV vaccination coverage in Colombia should be included to provide better context for the discussion.
Response: We appreciate the reviewer’s suggestion. In Colombia, equine influenza (EIV) vaccination is not mandatory at the national level for the general equine population. However, vaccination is required for the movement of horses. According to national regulations, all equines over six months of age must be vaccinated against EIV with products duly registered by the Colombian Agricultural Institute (ICA) in order to be eligible for transport. The vaccination is valid for one year. There are no national statistics on overall vaccination coverage, but it is generally understood that vaccination is mainly implemented in sport and competition horses and is not widely applied in non-competitive populations. This information has been added to the discussion to provide additional context regarding EIV exposure in Colombia. lines (339-342)
Minor comments for the authors.
- In the keyword, "influenza virus" should be written as "influenza A virus". Additionally, "vaccination" as "infection". The author should consider including "horses":
Response: Thank you for the comment. We have added the keywords: Vaccination, horses, Influenza A virus. ( linea 35)
- To ensure consistency, include the numerical figure beside the percentages where applicable. Response:
Response: Thank you, we have included it in line 169
- Line 75: Delete "other purpose" as it was repeated.
Response: Thank you, we replaced it with the term "use" in line 83
- Line 107: In this context, the correct word to use instead of "sensitized" is "coated."
Response: Thank you, we replaced it with "coated" in line 115
- Line 108-110: I believe this was performed using the manufacturer's reagents and chemicals. It will be important to refer to them as such.
Response: Thank you, we included it in line 119
- Line 117: change "viral" to "virus"
Response: Thank you, we made the change in line 125
- Line 233: Delete "positive"
Response: Thank you, we made the change in line 242
Reviewer 2 Report
Comments and Suggestions for Authors
The results of this study indicate that the positive rate of influenza antibodies in Colombian horses is relatively high. However, this positive rate includes the positive rate caused by infection and the positive rate of immunized vaccine horses within the validity period of the vaccine. Moreover, there is no significant difference in the positive rates between horses with respiratory symptoms (positive rates 31.4%, 18.2%) and horses without respiratory symptoms (positive rates 29.6%, 21.6%). These research results indicate that there are flaws in the research design: A. Only measuring NP antibodies cannot distinguish the positive rate of infection antibodies from the positive rate of vaccine immunity; B. The basic data of the sampled horses were incomplete (such as the time after immunization, the time after the appearance of respiratory symptoms, etc.). Therefore, if the following experiments, data and statistical results could be supplemented, this study would become more perfect.
- While determining NP antibodies, NS1 antibodies in serum should be measured simultaneously, or viral proteins should be determined by RT-PCR to distinguish between antibodies without antibodies and infection antibodies.
- Complete the basic data of each sampled horse, such as the time after vaccination and the time after the appearance of respiratory symptoms. Then, combine the data such as the production of antibodies about 10 days after vaccination and infection, the maintenance of infection antibodies for 6 months, and the half-life of vaccine antibodies being 6 months. This way, not only can the vaccine antibodies and infection antibodies be basically distinguished, but also the approximate time for booster immunization of the sampled horses can be proposed.
- Prevention and control measures for equine influenza should be proposed from three aspects: isolating infected horses (or horses with respiratory symptoms), cutting off the transmission route and protecting susceptible horses.
- The writing is not standardized. The hyphen between 7 and 10 on page 104 is in bold, and the sentence ends on pages 179, 191, 237, etc. should be solid full stops.
Author Response
The results of this study indicate that the positive rate of influenza antibodies in Colombian horses is relatively high. However, this positive rate includes the positive rate caused by infection and the positive rate of immunized vaccine horses within the validity period of the vaccine. Moreover, there is no significant difference in the positive rates between horses with respiratory symptoms (positive rates 31.4%, 18.2%) and horses without respiratory symptoms (positive rates 29.6%, 21.6%). These research results indicate that there are flaws in the research design: A. Only measuring NP antibodies cannot distinguish the positive rate of infection antibodies from the positive rate of vaccine immunity; B. The basic data of the sampled horses were incomplete (such as the time after immunization, the time after the appearance of respiratory symptoms, etc.). Therefore, if the following experiments, data and statistical results could be supplemented, this study would become more perfect.
- While determining NP antibodies, NS1 antibodies in serum should be measured simultaneously, or viral proteins should be determined by RT-PCR to distinguish between antibodies without antibodies and infection antibodies.
Response: In this study, we performed RT-PCR on the sampled animals as part of a broader research project, in which molecular detection of equine influenza virus (EIV) was one of the objectives. This analysis allowed us to identify positive cases of EIV, thereby confirming the presence of active infection in some of the evaluated horses (lines 368–370). This evidence, together with the time elapsed since the last vaccination (in most cases >6 months) and the presence of acute respiratory symptoms in all animals, supports the interpretation that the observed seropositivity is associated with recent viral exposure (lines 323–330).
We acknowledge that including assays targeting the NS1 protein could have further strengthened the differentiation between natural infection and vaccine-induced response, and we propose to consider this approach in future studies. This clarification and response are addressed in (lines 368–370).
- Complete the basic data of each sampled horse, such as the time after vaccination and the time after the appearance of respiratory symptoms. Then, combine the data such as the production of antibodies about 10 days after vaccination and infection, the maintenance of infection antibodies for 6 months, and the half-life of vaccine antibodies being 6 months. This way, not only can the vaccine antibodies and infection antibodies be basically distinguished, but also the approximate time for booster immunization of the sampled horses can be proposed.
Response: Thank you very much for your observation and comment. We have incorporated the analysis of the time elapsed after vaccination. All individuals were sampled between day 1 and day 5 after the onset of clinical signs. Following your suggestion, we analyzed ELISA positivity according to the time elapsed since the last vaccine administration, grouping horses as vaccinated either more than 6 months ago or within the last 6 months.Our analysis showed that 79.2% of seropositive horses had been vaccinated more than six months prior, while only 20.8% had been recently vaccinated (<6 months) (lines 173-174) and we include in the article. According to the literature, although vaccine-induced antibodies may persist for up to 12 months, virological protection begins to decline significantly as early as three months post-vaccination (Entenfellner et al., 2020; Colgate et al., 2022). This information has been included also in the Discussion (321-328 lines) sections.
- Prevention and control measures for equine influenza should be proposed from three aspects: Isolating infected horses (or horses with respiratory symptoms), cutting off the transmission route and protecting susceptible horses.
Response: Thank you very much for your observation. We included in lines (203-205) (328-333) prevention and control measures that arise from our findings.
- The writing is not standardized. The hyphen between 7 and 10 on page 104 is in bold, and the sentence ends on pages 179, 191, 237, etc. should be solid full stops.
Response: Thank you very much for your observation. We made the requested changes and adjusted the punctuation accordingly.
Reviewer 3 Report
Comments and Suggestions for Authors
This manuscript displays some minor mistakes marked in this uploaded file.

Author Response
This manuscript displays some minor mistakes marked in this uploaded file (Adjunto)
Response: Thank you, we included the recommendations.
Reviewer 4 Report
Comments and Suggestions for Authors
Dear Authors, please correct the marked sentences!

Author Response
Reviewer 4.
Dear Authors, please correct the marked sentences!
Response: Thank you, we included the recommendations.
Round 2
Reviewer 1 Report
Comments and Suggestions for Authors
In general, the authors addressed the basic queries raised. However, their response, particularly their rationale for affirming the presence of EIV H3N8 through PCR and NGS within the same study, does not adequately support the assumption that the detected influenza A antibodies were exclusively derived from H3N8. In scientific discourse, reliance on verifiable facts and evidence is paramount. The authors must present concrete evidence to confirm that 27% (n = 104) of the detected antibodies were indeed specific to H3N8. Employing a simple hemagglutination inhibition test or a more technical single radial hemolysis assay would be sufficient to clarify this issue.
The authors should be aware that the paper by Loroño-Pino et al. (2010; doi: 10.1136/vr.b5586), which they cited to support their use of ELISA, specifically examined antibodies for influenza A virus in multiple species, as well as antibodies against the West Nile virus. This differs from the current study, which focuses specifically on equine influenza.
In addition, the vaccination coverage explanation in the country should be summarised in the manuscript for the readers' benefit.
Comments on the Quality of English LanguageThe English language can be improved.
Author Response
Comment: In general, the authors addressed the basic queries raised. However, their response, particularly their rationale for affirming the presence of EIV H3N8 through PCR and NGS within the same study, does not adequately support the assumption that the detected influenza A antibodies were exclusively derived from H3N8. In scientific discourse, reliance on verifiable facts and evidence is paramount. The authors must present concrete evidence to confirm that 27% (n = 104) of the detected antibodies were indeed specific to H3N8. Employing a simple hemagglutination inhibition test or a more technical single radial hemolysis assay would be sufficient to clarify this issue.
Response: Thank you for your suggestion. We agree that confirming antibody specificity is essential for accurate interpretation of serological findings.As stated in the manuscript, the ELISA test employed (IDEXX AI Multi-Screen-Influenza A Ab ELISA Test®) detects antibodies against the nucleoprotein (NP) of influenza A viruses. While this method is robust for identifying prior exposure to influenza A, it does not distinguish between influenza A subtypes. Our interpretation that the detected antibodies are likely associated with EIV H3N8 is supported by two main points:
- Confirmed circulation of H3N8 in Colombia during the study period in a previous study involving symptomatic horses from Study 1: we previously confirmed EIV H3N8 infection via RT-PCR and next-generation sequencing (Gonzalez-Obando et al., Viruses 2024).
- Lack of reports of other influenza A subtypes in horses in Latin America to date, EIV outbreaks reported in South and Central America, including those in Argentina, Chile, Uruguay, Ecuador, and Colombia, have exclusively involved the H3N8 subtype in the last 30 años. We acknowledge that assays such as hemagglutination inhibition (HI) or single radial hemolysis (SRH) would allow more precise identification of subtype-specific antibodies. Unfortunately, due to logistical constraints, these tests could not be performed during the current study. We now emphasize this limitation in the revised discussion section (lines 378–391) and clarify that while H3N8 is the most likely subtype associated with seropositivity, further confirmation through HI or SRH testing would strengthen this conclusion.
Comment: The authors should be aware that the paper by Loroño-Pino et al. (2010; doi: 10.1136/vr.b5586), which they cited to support their use of ELISA, specifically examined antibodies for influenza A virus in multiple species, as well as antibodies against the West Nile virus. This differs from the current study, which focuses specifically on equine influenza.
Response: Thank you for your suggestion. We are aware that the study by Loroño-Pino et al. (2010) included multiple species and evaluated antibodies against both influenza A virus (IAV) and West Nile virus (WNV). However, our reference to this paper was made solely in relation to their findings in horses, which accounted for the majority of animals sampled (186 out of 266). In their study, IAV antibodies were detected in 25% of equine samples using a nucleoprotein-based ELISA, and these results were subsequently confirmed to be specific to the H3N8 subtype through hemagglutination inhibition (HI) and neuraminidase inhibition (NI) assays. Similarly, in our study, we observed a 27% seroprevalence of IAV antibodies in horses, and we also confirmed the presence of H3N8 in the same equine population through PCR analysis conducted on symptomatic animals, as described in the manuscript and in a previous publication (Gonzalez-Obando et al., Viruses, 2024). Therefore, while the overall scope of Loroño-Pino et al.’s work differs from ours, their findings in horses are directly relevant and support the use of ELISA for equine IAV surveillance. In both studies, the serological results were complemented by independent confirmation of the circulating subtype, strengthening the validity of the ELISA findings. We have clarified this point in the revised manuscript to ensure it is appropriately contextualized. We now clarified this limitation in the revised discussion section (lines 241–251)
Comment: In addition, the vaccination coverage explanation in the country should be summarised in the manuscript for the readers' benefit.
Response: Thank you for your suggestion. We agree that summarizing the vaccination coverage helps place our serological findings in better context. In Colombia, equine influenza vaccination isn’t mandatory and is mostly used in sport or high-value horses. According to data collected during our fieldwork, only 12.8% of the horses included in this study had received the vaccine.
This low coverage is particularly important, as it suggests that most of the antibodies we detected likely reflect natural exposure rather than a vaccine response. To our knowledge, this is also the first study in Colombia to report vaccination coverage in at-risk equine populations, which we believe adds valuable information for future prevention strategies. We now clarified this information in the revised Discussion section (lines 307–314)
Round 3
Reviewer 1 Report
Comments and Suggestions for Authors
The authors’ response to the query regarding the need for a confirmatory test for influenza A virus (IAV) antibodies detected is reasonable; however, it is based on certain assumptions. First, they posit that because PCR and NGS assays confirming H3N8 were conducted in the herd at some point in the past, the IAV antibodies detected in this study are likely linked to the previously reported H3N8 outbreak. In some cases, not all animals exposed to or vaccinated against a pathogen produce an antibody response due to various factors, including, but not limited to, health status, age, species etc. Second, lack of reports of EIV outbreaks without active surveillance may not accurately reflect the true situation regarding which virus is circulating in the country.
It is essential to emphasize that the absence of a confirmatory test for the detected IAV antibodies represents a significant limitation of the study, which must be addressed, regardless of whether this limitation was acknowledged in the manuscript. While I understand that logistical issues exist, we should not compromise the fundamentals of good science for any reason. As you have noted, conducting a confirmatory test is crucial to determine whether the detected IAV antibodies is specific for the H3 strain. The authors may be surprised to discover that some serum samples could test negative for H3.
The title and aim of the study, “Seroprevalence of equine influenza virus antibodies in horses from four localities in Colombia,” are explicit. As such, the data supporting these conclusions must be unequivocal.